

# Genome-wide identification and analysis of the *CNGC* gene family in maize

Lidong Hao[1] and Xiuli Qiao[2]

[1] College of Agriculture and Hydraulic Engineering, Suihua University, Suihua, HeiLongjiang province, China
[2] College of Food and Pharmaceutical Engineering, Suihua University, Suihua, HeiLongjiang province, China

## ABSTRACT

As one of the non-selective cation channel gene families, the cyclic nucleotide-gated channel (CNGC) gene family plays a vital role in plant physiological processes that are related to signal pathways, plant development, and environmental stresses. However, genome-wide identification and analysis of the *CNGC* gene family in maize has not yet been undertaken. In the present study, twelve *ZmCNGC* genes were identified in the maize genome, which were unevenly distributed on chromosomes 1, 2, 4, 5, 6, 7, and 8. They were classified into five major groups: Groups I, II, III, IVa, and IVb. Phylogenetic analysis showed that gramineous plant CNGC genes expanded unequally during evolution. Group IV CNGC genes emerged first, whereas Groups I and II appeared later. Prediction analysis of cis-acting regulatory elements showed that 137 putative cis-elements were related to hormone-response, abiotic stress, and organ development. Furthermore, 120 protein pairs were predicted to interact with the 12 ZmCNGC proteins and other maize proteins. The expression profiles of the ZmCNGC genes were expressed in tissue-specific patterns. These results provide important information that will increase our understanding of the *CNGC* gene family in maize and other plants.

## INTRODUCTION

Organism evolution has led to the formation of complex nutrient absorption and transport systems, including ion channels, ion pumps, and carriers. It has been previously shown that these systems respond to endogenous and abiotic stimuli (*Saand et al., 2015b*). The cyclic nucleotide-gated channel (CNGC) is a $Ca^{2+}$-permeable cation transport channel, and it has been suggested that it is one of the fundamental mechanisms in organism systems (*Yuen & Christopher, 2013*; *Nawaz et al., 2014*). Secondary messengers, such as cyclic nucleotide monophosphates (3′,5′-cAMP and 3′,5′-cGMP) and $Ca^{2+}$/calmodulin (CaM), can regulate CNGCs by acting as molecular switches. The CNGCs are activated by directly binding cyclic nucleotides and are inhibited when CaM binds to the CaM binding domain (*Saand et al., 2015b*; *Borsics et al., 2007*; *Defalco et al., 2016*; *Kaplan, Sherman & Fromm, 2007*).

In plants, CNGCs are composed of six transmembrane (TM) domains and one pore region between the fifth and sixth TM domains. The cyclic nucleotide-binding domain (CNBD) is a highly conserved region and has a phosphate-binding cassette (PBC) and a hinge region (*Saand et al., 2015b*). Unfortunately, although the existence of these domains is

Corresponding authors
Lidong Hao, haolidong1987@163.com
Xiuli Qiao, shxynxhld@163.com

necessary for CNGC function, they cannot be used to identify a CNGC protein because other ion transporters such as potassium AKT/KAT channels (Shaker type) also contain both a CNBD and a TM domain (*Su et al., 2001*; *Chérel, 2004*). However, previous studies have proposed that a plant CNGC-specific motif, [LIMV0]-X(2)-[GSANCR]-X-[FVIYASCL]-X-G-X(0,1)-X(0,1)-[EDAQGH]-L-[LIVFA]-X-[WRCMLS0]-X-[LMSIQAFT0]-X(7,37)-[SAC]-X(9)-[VTIALMS]-X(0,1)-[EQDN]-[AGSVT]-[FYL]-X-[LIVF], in the PBC and hinge region within the CNBD of CNGC proteins only exists in plant CNGCs and does not occur in other ion transporters (*Zelman, Dawe & Berkowitz, 2013*; *Saand et al., 2015b*).

The first plant CNGC was identified in *Hordeum vulgare* and named *HvCBT1* (*Schuurink et al., 1998*). The use of bioinformatics tools has led to the identification of the *CNGC* gene family members in *Arabidopsis (20)*, rice (16) and other plants (18 in tomato and 26 in *Brassica oleracea*) (*Bridges, Fraser & Moorhead, 2005*; *Nawaz et al., 2014*; *Ward, Mäser & Schroeder, 2009*; *Zelman et al., 2012*; *Saand et al., 2015a*; *Chen et al., 2015*; *Zelman, Dawe & Berkowitz, 2013*; *Guo et al., 2017*; *Kakar et al., 2017*).

Previous studies have shown that CNGCs are key components of plant development. Most *CNGC*s have been characterized by genetic methods. They have been shown to play vital roles related to plant physiological and molecular functions, such as multiple physiological processes involved in signal pathways, plant development, and responses to environmental stresses. For example, *Arabidopsis CNGC7* and *CNGC8* are essential for male reproductive fertility (*Tunc-Ozdemir et al., 2013a*); *CNGC16* and *CNGC18* participate in pollen development (*Tunc-Ozdemir et al., 2013b*; *Frietsch et al., 2007*; *Gao et al., 2016*); *AtCNGC2* is involved in jasmonic acid induced apoplastic $Ca^{2+}$ influx in epidermal cells (*Lu et al., 2015*; *Wang et al., 2017*); and *Arabidopsis CNGC6*, *CNGC19*, and *CNGC20* are involved in abiotic stress response (*Kugler et al., 2009*; *Gao et al., 2012*). *Arabidopsis* CNGC structures have six TM domains and a pore domain. They also possess a cyclic nucleotide-binding domain and CaM-binding domain in the C-terminus. These various domains have diverse functions (*Talke et al., 2003*; *Chin, Moeder & Yoshioka, 2009a*; *Hua et al., 2003*; *Köhler & Neuhaus, 2000*). For example, *AtCNGC2* plays a key role in stress signaling pathways, including changes to the cytosolic free $Ca^{2+}$ in *Arabidopsis*. In contrast, *CNGC4* is permeable to $K^+$ and $Na^+$, and is activated by both cGMP and cAMP (*Balague, 2003*; *Ali et al., 2007*).

In recent years, efforts had been made to study the *CNGC* gene family in plants. However, there have been few studies on the maize *CNGC* gene family, even though maize is an important food crop and a source of industrial materials worldwide. This study used maize genome-wide sequence information, research information on *Arabidopsis* and rice CNGC families, and comprehensive bioinformatics analysis techniques to conduct a genome-wide identification of CNGCs in maize. To the best of our knowledge, this is the first systematic study of *CNGC* genes in maize and provides the basis for future research on the *ZmCNGC* gene family.

## MATERIALS AND METHODS

### Identification of *CNGC* genes in the maize genome

A total of 20 *Arabidopsis* and 16 rice *CNGC* protein sequences were retrieved from the *Arabidopsis* Information Resource (TAIR10) database (http://www.arabidopsis.org/) and the Rice Genome Annotation Project (RGAP) database (http://rice.plantbiology.msu.edu/), respectively. This information was then used to identify the *CNGC* genes in maize. Two methods were utilized to search the maize protein sequences. One used a Hidden Markov Model (HMM) to search against maize protein sequences and the other used the local BLASTP method with a threshold *e*-value <1e−5. After the searches were conducted, a manual correction was performed to remove any redundancy and proteins without PBC and hinge regions within the CNBD of the CNGC proteins. To further confirm whether the ZmCNGC proteins contained the CNBD domain, the putative ZmCNGC protein sequences were submitted to the SMART (*Letunic & Bork, 2018*) and NCBI-CDD databases (*Marchler-Bauer et al., 2017*). The proteins without CNBD domains or with amino acid (aa) numbers below 200 were removed and the ZmCNGCs confirmed. Another 11 gramineae plant CNGCs were identified by applying the same method as that described above.

The PI (theoretical isoelectric point), MW (molecular weight), and GRAVY (grand average of hydropathy) of the ZmCNGCs were predicted by ExPASy (*Artimo et al., 2012*). CELLO v.2.5 software was used to predict the subcellular location of the ZmCNGCs (*Yu, Lin & Hwang, 2010*). Information about the chromosome distribution of ZmCNGCs and genetic sequences, such as DNA sequences, CDS, cDNA, and up-stream 1500 base pair (bp) DNA sequences obtained from a BLASTN search of the Ensembl Plant database (*Bolser et al., 2016*).

### Multiple alignments, phylogenetic analysis and gene duplication analysis

Multiple sequence alignments were performed using the T-COFFEE method (*Di Tommaso et al., 2011*) and visualized by ESPript using the default program setting (*Robert & Gouet, 2014*). A maximum likelihood (ML) phylogenetic tree was constructed using the MEGA 7 software program with 1,000 bootstrap replications and the Jones-Taylor-Thornton model (*Kumar, Stecher & Tamura, 2016*). To further validate the accuracy of the ML tree, an un-rooted phylogenetic tree was constructed with 1,000 bootstrap replications using the MEGA 7 software program and was based on full-length protein sequence alignments. The ML analysis of 12 gramineae plants was performed using the IQTree program with the LG + G4 and state frequencies were determined from the amino acid matrix and other default parameters (*Lam-Tung et al., 2015*). The tree was visualized by Evolview online (*He et al., 2016*). Segmental duplication between maize genes and the synteny block between maize and *Sorghum*, and rice and *Brachypodium* were obtained from the Plant Genome Duplication database (*Lee et al., 2013*). The substitution rates (Ka/Ks) for duplication events were calculated by the DnaSP v5 software program (*Librado & Rozas, 2009*), and the divergence times (Mya) were calculated using the formula: Mya = Ks $/2\lambda \times 10^{-6}$, where $\lambda = 6.5 \times 10^{-9}$ (*Lynch & Conery, 2000*).

## Gene structure and conserved motif analyses

The gene structures (exon-intron) of the *ZmCNGC* genes were determined by the Gene Structure Display Server (*Hu et al., 2015*) using the CDS and DNA sequences from the *ZmCNGC* genes. The conserved motifs of the ZmCNGC proteins were subjected to the MEME Suite web server (*Bailey et al., 2015*) with the maximum number of motifs set at 10 and an optimum width range for the motifs of 6–200 aa.

## Cis-acting regulatory elements and the prediction of protein–protein interaction in ZmCNGCs

The up-stream 1,500 bp DNA sequences in the *ZmCNGC* genes were used to locate cis-acting regulatory elements by the 'Signal Scan Search' programs in the NEW PLACE database (*Higo et al., 1999*). The interactions between ZmCNGCs and other maize proteins were predicted using the STRING 10 online program (*Szklarczyk et al., 2017*) and visualized by the Cytoscape v3.4.0 software program (*Shannon et al., 2003*).

## *ZmCNGC* gene expression profiles and network interaction analysis

To understand the expression of *ZmCNGC* genes in different tissues, two high-throughput datasets for of maize were obtained from the Expression Atlas datasets (https://www.ebi.ac.uk/gxa/home/) under accession numbers E-MTAB-3826 and E-MTAB-439. These data were used to analyze the expression of *ZmCNGC* genes in six different tissues (ear, embryo, endosperm, pollen, root, and tassel) and at different developmental stages (embryo, endosperm, and seed). The FPKM values were used to calculate the *ZmCNGC* genes expressions and these were visualized by OmicShare tools, which is a free online platform for data analysis (http://www.omicshare.com/tools).

# RESULTS

## Identification of the *CNGC* genes in maize

A total of 20 *Arabidopsis* and 16 rice CNGC protein sequences which had been BLAST aligned against maize protein sequences, were used to obtain an overview of the *CNGC* genes in maize. After BLAST alignment, 18 putative *ZmCNGC* genes were identified in the maize genome. The 18 putative *ZmCNGC* genes were confirmed by using SMART and NCBI CDD to determine whether they contained the CNGC-specific domains (CNBD and TM). After removing redundant genes, 12 *ZmCNGC* genes were finally identified, which was lower than the number of *CNGC* genes in rice and *Arabidopsis* (*Paterson, Bowers & Chapman, 2004*; *Yu et al., 2005*). The predicted *ZmCNGC* genes were designated as ZmCNGC1 to ZmCNGC12 based on their family classification (Table 1). To further confirm the existence of the ZmCNGCs, we identified all the expressed sequence tags (ESTs) that had aligned to the *ZmCNGC* genes by using the BLASTN program provided by the NCBI. The results demonstrated that only ZmCNGC3 had showed no EST hits, whereas the other ZmCNGCs had more than 13 representative matches to ESTs. Five of them were located on chromosome 5, whereas the others were unevenly located on chromosomes 1, 2, 4, 6, 7, and 8. The characteristic features of these 12 *ZmCNGC* genes are listed in Table 1. The ZmCNGC protein lengths ranged from 326 to 745 aa with an average

Hao and Qiao (2018), *PeerJ*, DOI 10.7717/peerj.5816

**Table 1  Characteristic features of *ZmCNGC* genes in maize.**

| Group | Gene ID | Gene name | Chr[a] | Start[b] | end[c] | Length (aa)[d] | MW (Da)[e] | pI[f] | GRAVY[g] | Localization[h] | EST[i] |
|-------|---------|-----------|--------|----------|--------|----------------|------------|-------|----------|-----------------|--------|
| I | ZmCNGC1 | GRMZM2G148118 | 4 | 230194909 | 230205383 | 701 | 79914.66 | 9.18 | −0.046 | PlasmaMembrane | 17 |
| | ZmCNGC2 | GRMZM2G129375 | 6 | 109824359 | 109825921 | 626 | 38405.22 | 9.17 | −0.484 | PlasmaMembrane | 23 |
| | ZmCNGC3 | GRMZM2G066269 | 4 | 230382717 | 230384596 | 329 | 38632.18 | 9.59 | −0.52 | Nuclear | 0 |
| II | ZmCNGC4 | GRMZM2G023037 | 2 | 5966501 | 5978989 | 723 | 82856.52 | 8.92 | −0.114 | PlasmaMembrane | 29 |
| | ZmCNGC5 | GRMZM2G077828 | 5 | 17453069 | 17455739 | 699 | 80110.81 | 9.42 | −0.091 | PlasmaMembrane | 15 |
| III | ZmCNGC6 | GRMZM2G005791 | 5 | 218834435 | 218840917 | 700 | 80057.39 | 8.97 | −0.078 | PlasmaMembrane | 31 |
| | ZmCNGC7 | GRMZM2G068904 | 5 | 191609046 | 191611784 | 689 | 80038.95 | 9.8 | −0.117 | PlasmaMembrane | 15 |
| | ZmCNGC8 | GRMZM2G135651 | 7 | 150652512 | 150657060 | 739 | 85523.26 | 9.33 | −0.148 | PlasmaMembrane | 61 |
| IVa | ZmCNGC9 | GRMZM2G141642 | 5 | 217119224 | 217125465 | 463 | 53303.64 | 9.48 | −0.059 | PlasmaMembrane | 13 |
| IVb | ZmCNGC10 | GRMZM5G858887 | 5 | 6938133 | 6943695 | 745 | 83672.62 | 9.46 | 0.081 | PlasmaMembrane | 84 |
| | ZmCNGC11 | GRMZM2G074317 | 1 | 283401507 | 283408034 | 730 | 81440.82 | 9.43 | 0.067 | PlasmaMembrane | 74 |
| | ZmCNGC12 | GRMZM2G090528 | 8 | 177244867 | 177247281 | 505 | 57042.47 | 9.75 | 0.014 | PlasmaMembrane | 48 |

**Notes.**

[a] chromosome location.
[b] star of gene location.
[c] end of gene location.
[d] the length of ZmCNGC proteins.
[e] the molecular weight of ZmCNGC proteins.
[f] the Grand average of hydropathicity of ZmCNGC proteins.
[g] the prediction the ZmCNGCs subcellular location.
[h] expressed sequence tags blast results.

of 612 aa. The molecular weights of these proteins ranged from 38.63 kDa (ZmCNGC2) to 85.52 kDa (ZmCNGC8) and the calculated pI values ranged from 8.92 (ZmCNGC4) to 9.75 (ZmCNGC12). Subcellular localization prediction analysis showed that all the ZmCNGCs were localized in the plasma membrane except for ZmCNGC3 which was localized in the nuclear fraction. These results are similar to *Arabidopsis* (*Lemtiri-Chlieh & Berkowitz, 2004*).

## Multiple alignments of maize CNGCs and potassium AKT/KAT channel genes

Many ion transporters other than CNGCs also have a CNBD in the C-terminus and a hexa-TM in the N-terminus. For example, potassium AKT/KAT channels (Shaker type) also contain a CNBD and a TM domain. All AKT/ KAT-type channels consist of six TM regions with one P region (*Su et al., 2001*). Therefore, 11 maize AKT/KAT proteins were obtained from the NCBI resource and aligned with the 12 ZmCNGC protein sequences (File S1). The results showed that the proteins were highly conserved and that all of them contained six TM domains (S1-S6) and a pore region. The PBC and hinge domain were also highly conserved. A ML phylogenetic tree indicated that maize CNGC and AKT/KAT-type channel genes were clustered into two separate sections (File S1).

The CNBD is a gene structural feature element in plant CNGCs that contains the PBC and hinge region (*Diller et al., 2001*). Figure 1 illustrates that within the maize PBCs, a conserved phenylalanine (F), a stabilizing glycine (G) and an acidic residue (D or E), and two aliphatic leucines (L) were 100% conserved inside the PBCs. Additionally, aromatic phenylalanine (F) and leucine (L) were 100% conserved within the hinge region. These two conserved regions occurred between the CNBD and CaMBD regions. Based on corresponding alignments with other plants, a stringent motif (L-X(2)-G-[ED]-ELL-[TSG]-W-[ACY]-L-X(10,20)-[SA]-X-T-X(7)-[EQ]-[AG]-F-X-L) which was recognized in the 12 maize CNGCs that included the PBC and hinge domain, was found to be consistent with other plant species (*Saand et al., 2015b*; *Nawaz et al., 2014*). The maize, rice, and *Arabidopsis* CNGCs were also aligned. The results showed that no positions were specific to the maize CNGC consensus, which suggested that the PBCs and hinge domain were highly conserved among plants (File S2).

## Phylogenetic and duplication analyses of ZmCNGCs

Large phylogenetic trees with minimal homologous characters increase the likelihood of confounding relationships between different species. Therefore, we constructed a ML phylogenetic tree based on the alignment of CNGC proteins from 12 gramineae plants. The tree included 24 CNGC proteins in *Aegilops tauschii*, 16 in *Brachypodium distachyon*, 20 in *Hordeum vulgare*, 23 in *Leersia perrieri*, 21 in *Nicotiana attenuata*, 16 in *Oryza sativa*, 28 in *Setaria italic*, 13 in *Sorghum bicolor*, 79 in *Triticum aestivum*, 21 in *Triticum urartu*, and 12 maize CNGCs identified by this study (Table S1). The phylogenetic tree (File S4), was used to cluster these plant CNGC proteins into six groups, which were named Groups I, II, III, IV, IVa, and IVb, all of which had significant bootstrap values. The results showed that the CNGC proteins from *B. distachyon*, *O. sativa*, *S. bicolor*, and maize did not cluster

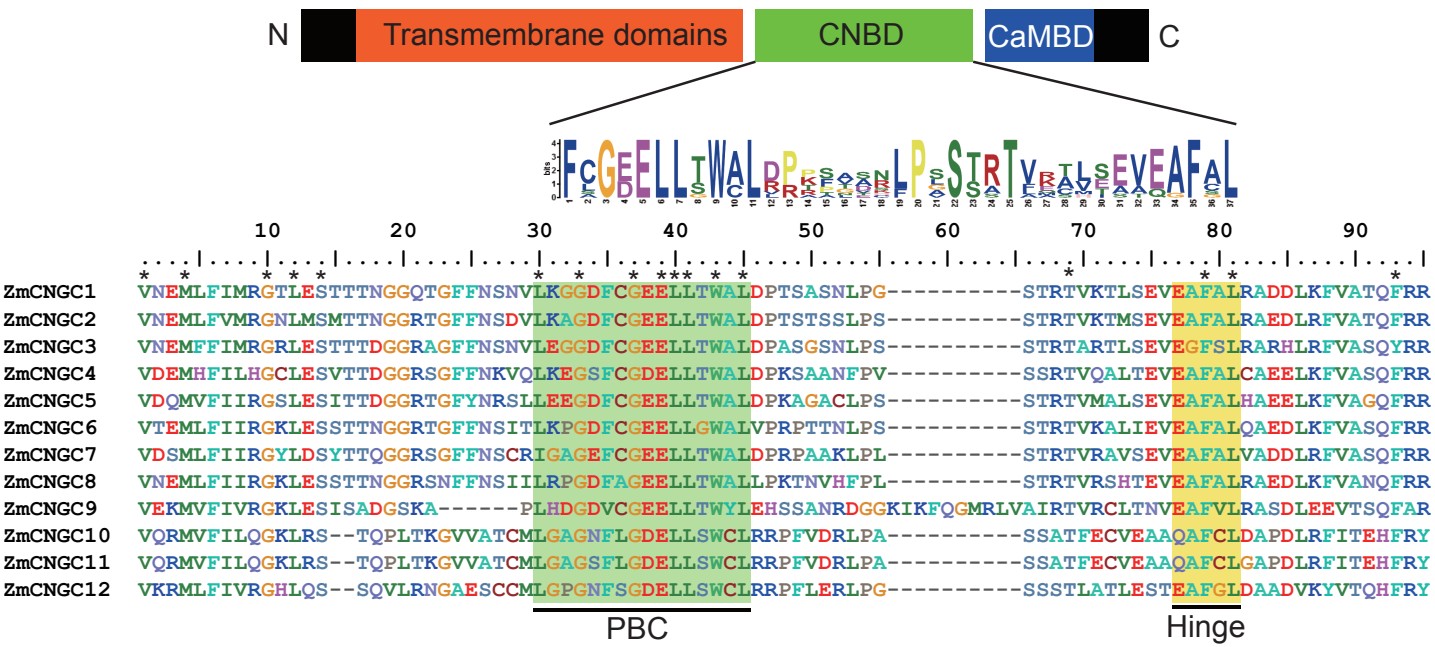

**Figure 1** **The maize CNGC-specific motif spans the putative PBC and the hinge within the CNBD of the 12 ZmCNGCs.** The diagram at the top represents three regions of plant CNGCs: the six transmembrane domains (TM), a CNBD containing a PBC and the hinge, and CaMBD. The maize CNGC-specific amino acid motif is shown below the cartoon. The asterisk indicate 100% identity among the 12 ZmCNGCs. Below is the alignment of CNBD domains of 12 ZmCNGCs. Residues in blue and yellow highlighted indicate the PBC and hinge domain, respectively.

in Group IV, and the *N. attenuata* CNGCs did not cluster in Group I. Additionally, the number in each group was unevenly distributed. Group IV was the largest with 86 genes, followed by 69 in Group III, 44 in Group II, 33 in Group IVb, 25 in Group I, and 16 in Group IVa. These data demonstrate that gramineae plant CNGC gene expansion appeared to have occurred unequally during evolution.

To better understand the evolutionary relationship among CNGC proteins, a ML phylogenetic tree was created based on the full-length protein alignments of 12 *ZmCNGC*s, 19 *AtCNGCs* and 16 *OsCNGCs* (*Nawaz et al., 2014*; *Maser et al., 2001*). The phylogenetic tree shows that the 47 CNGC proteins could be classified into five groups with significant bootstrap values (Fig. 2). These were named Groups I, II, III, IVa, and IVb. This was consistent with what has been previously reported for flowering plant CNGCs (*Saand et al., 2015b*). Group I contained three maize *CNGC* genes (*ZmCNGC1, ZmCNGC2,* and *ZmCNGC3*), five *Arabidopsis*, and two rice *CNGC* genes; Group II contained two maize (*ZmCNGC4* and *ZmCNGC5*), five *Arabidopsis*, and three rice *CNGC* genes. Group III contained three maize *CNGC* genes (*ZmCNGC6, ZmCNGC7,* and *ZmCNGC8*); and Group IV contained five rice and four *Arabidopsis CNGC* genes. The maize genes in Group IV were separated into two groups, Group IVa contained *ZmCNGC9* and Group IVb contained three (*ZmCNGC10, ZmCNGC11,* and *ZmCNGC12*).

The phylogenetic tree also showed that the maize *CNGC* genes could also be grouped into five groups (Fig. 3A). Consistent with other plant *CNGC* genes, Group IVa contained
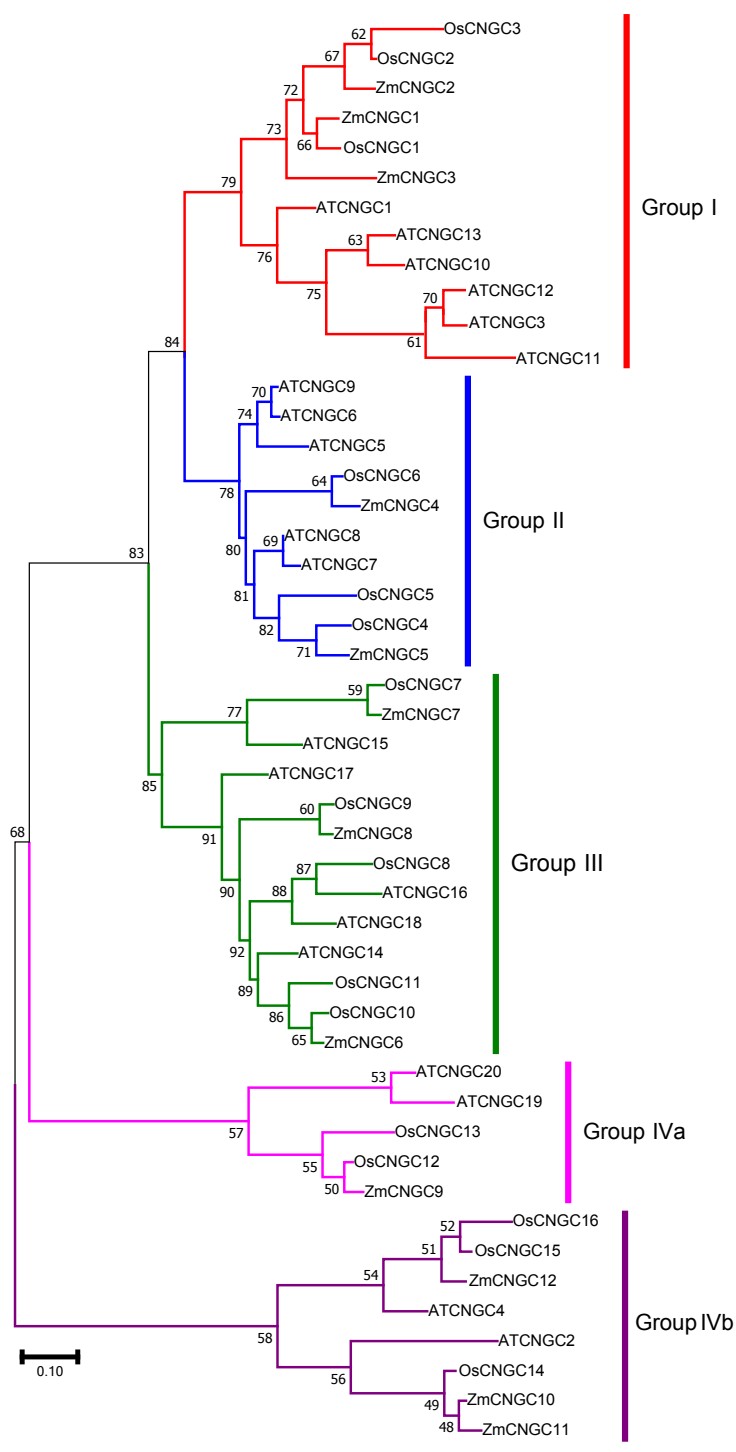

**Figure 2 Phylogenetic relationships among the ZmCNGCs, OsCNGCs and AtCNGCs.** The multiple alignment was performed by ClustalX program. MEGA 7.0 was used to create maximum likelihood (ML) under the Jones-Taylor-Thornton (JTT) model. The tree was constructed with 1,000 bootstrap replications. Each group identified is indicated on the right.

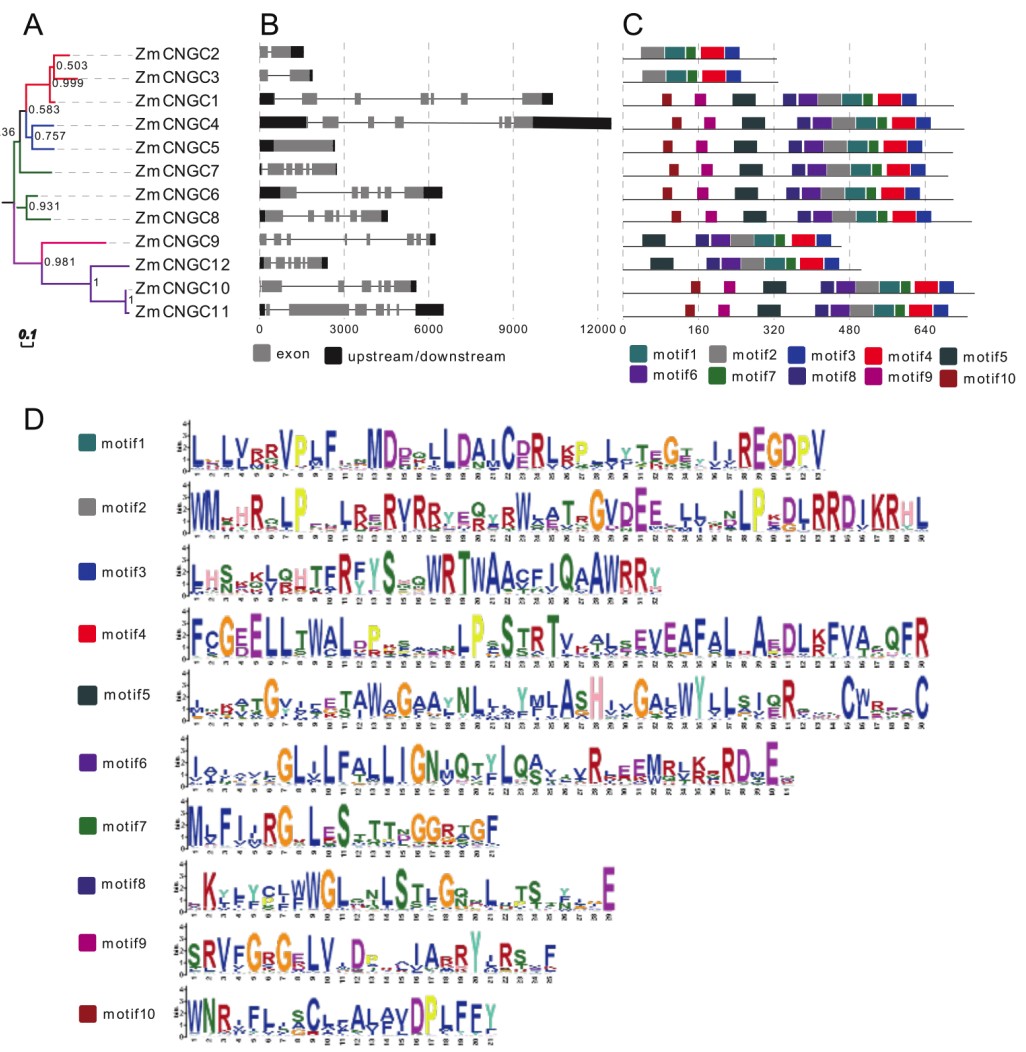

**Figure 3** **Phylogenetic relationships (A), motif compositions (B), gene structure (C), and (D) motif logo of ZmCNGCs.** The maximum likelihood (ML) tree under the Jones-Taylor-Thornton (JTT) model was constructed with 1,000 bootstrap replications using MEGA7 based on the full-length protein sequence. The exon-intron structures of these genes were graphically displayed by the Gene Structure Display Server using the CDS and genome sequence of ZmCNGC genes. The protein sequences of ZmCNGC genes were used to predict the conserved motifs by using the MEME Suite web server.

only one or two gene members and formed the smallest group (*Saand et al., 2015b*). The results suggested that during the evolution of CNGCs, Group IV *CNGC* genes may have emerged first whereas Groups I and II *CNGC* genes appeared later. Furthermore, the tree topology produced by the neighbor joining analyses was the same as the ML tree in Fig. 2, and it contained all the groups (File S3).

Additionally, only one segmental duplication gene pair, *ZmCNGC10-ZmCNGC11*, was formed in the maize genome (Table 2). The evolutionary process between maize CNGCs and other gramineae plants was further explored by investigating the genome synteny among *Sorghum*, rice, and *Brachypodium*. The results showed that there were two, two

**Table 2  The Ka/Ks ratios and estimated divergence time for orthologous CNGC proteins between maize and other Gramineae plants.**

| Gene ID | Gene ID | Ka | Ks | Ka/Ks | Mya |
|---------|---------|------|------|--------|-----|
| ZmCNGC11 | ZmCNGC10 | 0.0258 | 0.1829 | 0.141061 | 14.06923 |
| ZmCNGC5 | Sobic.001G155100 | 0.5298 | 0.8917 | 0.594146 | 68.59231 |
| ZmCNGC12 | Sobic.009G188800 | 0.095 | 0.4878 | 0.194752 | 37.52308 |
| ZmCNGC12 | Sobic.003G317700 | 0.0306 | 0.0993 | 0.308157 | 7.638462 |
| ZmCNGC5 | LOC_Os03g44440 | 0.7415 | 1.2253 | 0.605158 | 94.25385 |
| ZmCNGC12 | LOC_Os05g42250 | 0.1187 | 0.4278 | 0.277466 | 32.90769 |
| ZmCNGC12 | LOC_Os01g57370 | 0.073 | 0.3017 | 0.241962 | 23.20769 |
| ZmCNGC5 | Bradi1g13740 | 0.7912 | 1.1721 | 0.675028 | 90.16154 |

and one *ZmCNGC* genes that showed syntenic bias towards particular *Sorghum*, rice, and *Brachypodium* chromosomes, respectively (Table 2). In addition, Ka/Ks was used to evaluate their specific positions under positive selection pressure after duplication (*Mayrose et al., 2007*). Ka/Ks values that =1, <1 or >1 indicate neutral, purifying, and positive selection, respectively (*Lynch & Conery, 2000*). The Ka/Ks value of each gene pair was calculated and the Ka/Ks value for all gene pairs was less than 1, which suggested that these genes had evolved under strong purifying selection. Furthermore, the results indicated that the divergence of maize CNGCs from other gramineae plants did not occur evenly.

### *ZmCNGC* gene structures and the conserved motif analyses

Gene structure analysis could improve our understanding of gene function and evolution. The number of introns ranged from 0 to 7 (Fig. 3B), which was different from rice and *Arabidopsis* CNGCs. In rice, *the OsCNGC* s introns numbers ranged from 1 to 11, whereas *Arabidopsis* CNGCs ranged from 4 to 10 (*Nawaz et al., 2014*). The Group IVa and IVb ZmCNGCs had distinct gene structures compared to those of the other groups, with more introns at different phases and lengths, which is consistent with most flowering plant species Group IV *CNGC* genes (*Saand et al., 2015b*).

The motif-based recognition of proteins can improve understanding of their the evolutionary history (*Seoighe & Gehring, 2004*). Ten putative motifs were characterized and named as motif1 to motif10 in the ZmCNGCs. The relative positions of the motifs in the five groups were found to have various patterns (Figs. 3C and 3D), for example, all the ZmCNGCs contained motif1, motif2, motif3, and motif4, which meant they presented typical ZmCNGC domains. Motif3 was a combination of the calmodulin binding (CaMB) domain and the IQ domain (QWRTWAA[CV]FIQ[AL]AW[RH]RY), and motif4 was the cyclic nucleotide-binding (CNB) domain, which was located in the C-terminal. Furthermore, motif10, 9, 5, 8 (or 6), 2, and 1 were the motif logos of the transmembrane domains. They represented the S1, S2, S3, S4, S5, and S6 regions of the transmembrane domain in the N-terminal. Motif7 represented the ion transport protein (ITP) domain. In addition, 10 of the ZmCNGCs, but not *ZmCNGC2* and *ZmCNGC3*, possessed motif 5 and 6 that are associated with ion transport (*Nawaz et al., 2014*). Although other motifs

have not been reported in plants or animals, they undertake important functions within the organism.

## Prediction of cis-acting regulatory elements and protein-protein interactions that involve ZmCNGC proteins

Cis-acting regulatory elements are important molecular switches that are associated with the transcriptional regulation of genes when environmental stresses are encountered (*Nakashima, Ito & Yamaguchi-Shinozaki, 2009*). To better understand the possible biological processes undertaken by the ZmCNGCs involved, 1.5 kb sequences upstream of the *ZmCNGC* gene genomic sequences were used to identify cis-regulatory elements. A total of 137 different putative cis-elements were found to be associated with the identified *ZmCNGC* genes and only 12, including CACTFTPPCA1, EBOXBNNAPA, DOFCOREZM, MYCCONSENSUSAT, CAATBOX1, GTGANTG10, WRKY71OS, GT1CONSENSUS, ROOTMOTIFTAPOX1, POLLEN1LELAT52, MYBCORE, and OSE2ROOTNODULE, were in the promoter region of the *ZmCNGC* genes (Table S2), which was highly consistent with rice CNGCs. Additionally, five cis-elements were gene-specific. ACGTCBOX, TATABOX3, CTRMCAMV35S, and HDZIP2ATATHB2 were unique to *ZmCNGC6*, *ZmCNGC7*, *ZmCNGC8*, and *ZmCNGC11*, respectively. Additionally, some cis-elements were involved in responses to different abiotic/biotic stimuli, including hormones (e.g., abscisic acid, auxin, ethylene), stress (e.g., drought, temperatures, disease) and development (e.g., mesophyll specific, tissue specific) responses, which indicated that these *ZmCNGC* genes might be involved in regulating diverse stress responses.

The ZmCNGC protein-protein interactions were predicted in order to gain a better understanding of ZmCNGC protein roles in the plant. A total of 120 protein pairs were predicted to interact between the 12 ZmCNGC proteins and the other 11 maize proteins, and ZmCNGC8 was found to interact with *ZmCNGC10* by prediction analysis (Fig. 4 and Table S3). We also used genes in *Arabidopsis* that were homologous to the *ZmCNGC* genes to predict the protein-protein interactions. The results showed that three were validated by the experimental data (Table S3).

## Expression profiles of *ZmCNGC* genes in different tissues

The physiological function of *ZmCNGC* genes was investigated using transcriptome sequencing which evaluated the tissue-specific expression levels of *ZmCNGC* genes in different tissues (see Materials and Methods section). The expression levels among the *ZmCNGC* genes are shown in Fig. 5A and Table S4 suggested that their expression levels were tissue-specific. For example, *ZmCNGC5* was specifically expressed in pollen compared to other tissues, which implied that it played a particular role in pollen physiological development, whereas *ZmCNGC2*, *ZmCNGC4*, *ZmCNGC6*, and *ZmCNGC8* had higher expression levels in the roots than in the other tissues, which suggested that they had important roles in root growth and development. All Group IVb *ZmCNGC* genes, including *ZmCNGC10*, *ZmCNGC11*, and *ZmCNGC12*, were relatively highly expressed in the embryo, which implied that these genes played crucial roles in the growth and development of the maize embryo.

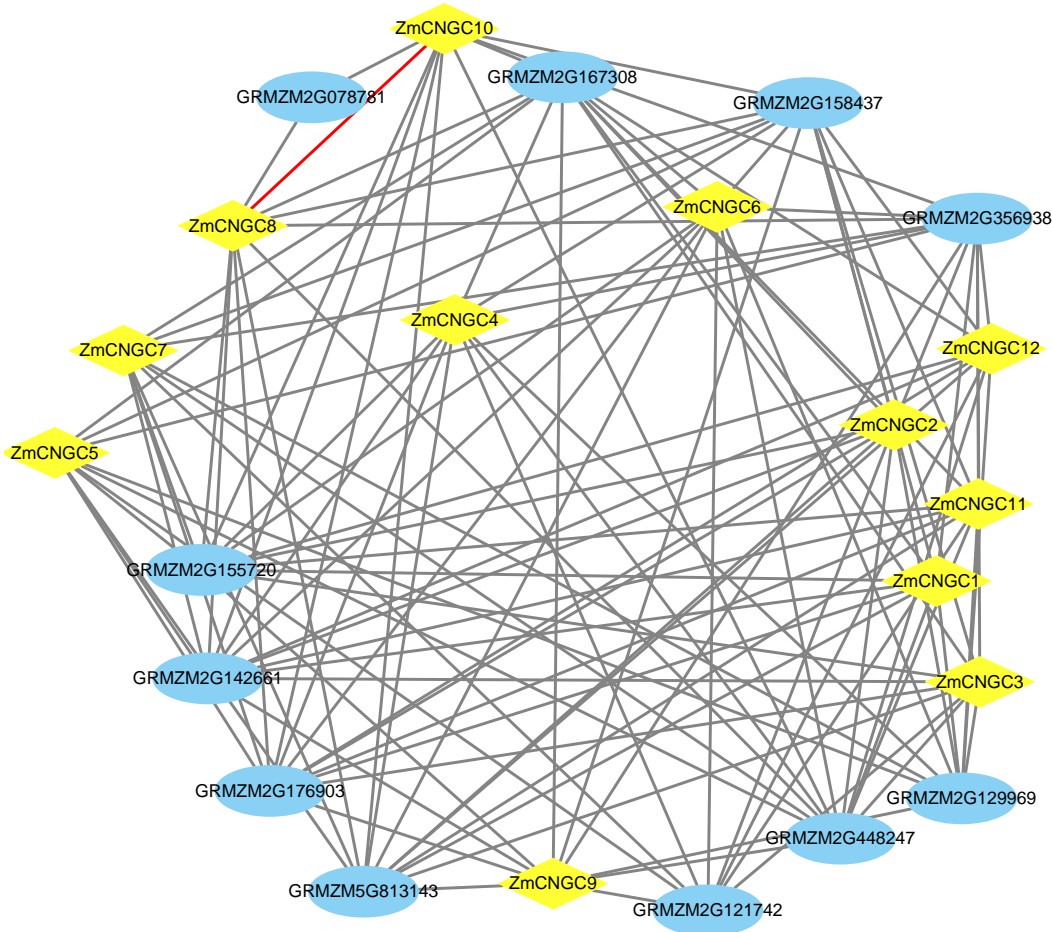

**Figure 4** **The protein–protein interaction network of ZmCNGC genes in maize.** The protein–protein interaction network were constructed based on STRING database.

We also evaluated the expression of several *ZmCNGC* genes in the embryo, endosperm, and seed for several days after pollination. *ZmCNGC3*, *ZmCNGC5* and *ZmCNGC7* were not detected or had no expression in any of the tissues. However, *ZmCNGC5* was expressed in pollen (Figs. 5A, 5B, 5C, and 5D). Cis-acting regulatory elements analysis showed that only *ZmCNGC5* and *ZmCNGC7* did not contain CANBNNAPA, which is the element required for embryo and endosperm-specific transcription (*Ellerström et al., 1996*). This might be the reason why they did not show any expression in these tissues. The embryo specific-expression gene *ZmCNGC10* gradually increased over time in the embryo (Fig. 5B). *ZmCNGC8* was highly expressed in the embryo, endosperm, and seeds after pollination, and the Group IVb gene showed a similar expression pattern.

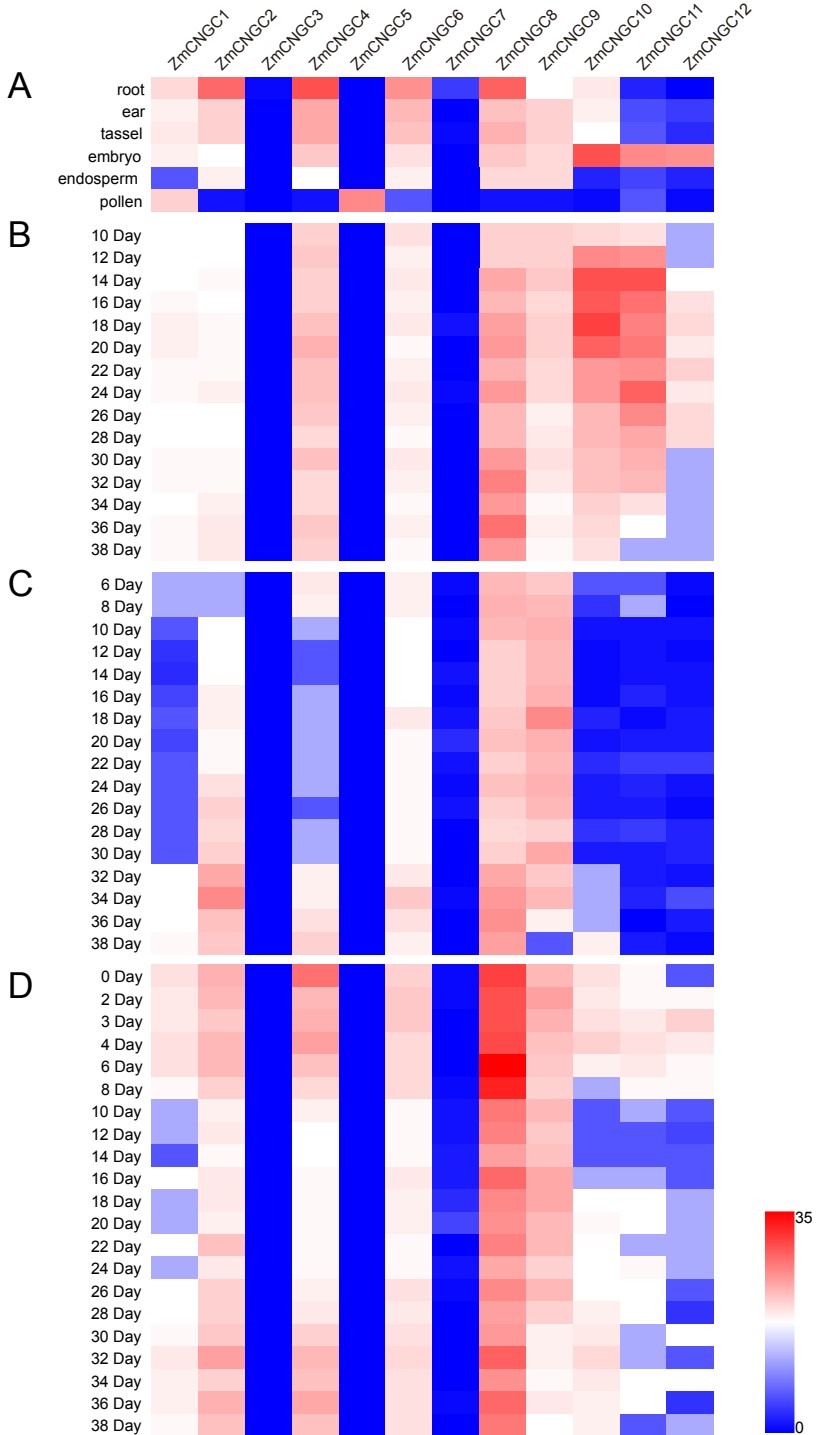

**Figure 5** **Expression profiles of ZmCNGCs in six different tissues and different development stages in embryo, endosperm and seed after pollination, respectively.** (A) Expression profile in six different tissues, including ear, embryo, endosperm, pollen, root and tassel. Expression profile in (B) embryo, (C) endosperm and (D) seed of some days after pollination.

## DISCUSSION

### Features and evolution of plant CNGC family genes

Plant CNGC family genes are characterized by the presence of a CNBD in the C-terminal and a hexa-TM in the N-terminal (*Saand et al., 2015b*). A total of 12 *ZmCNGC* genes were identified after a BLAST search against the maize genome protein sequences. Among them, ZmCNGC3 showed no ESTs and was not expressed in all the tissues, which suggested that it was a non-expressed pseudogene. In *Arabidopsis*, AtCNGC16 (AT3G48010) had no EST alignments and might also be a pseudogene (*Maser et al., 2001*). Many other ion transporters also possess these domains. For example, potassium AKT/KAT channels (Shaker type) contain both a CNBD domain and a TM domain. All AKT/ KAT-type channels consist of six transmembrane (TM) regions with one P region (*Su et al., 2001*). The ML phylogenetic tree showed that the maize CNGC and the AKT/KAT-type channel genes were clustered into two separate sections (File S1). Previous studies have shown that the CNGC-specific motif with a PBC and a hinge domain (L-X(2)-G-[ED]-ELL-[TSG]-W-[ACY]-L-X(10,20)-[SA]-X-T-X(7)-[EQ]-[AG]-F-X-L) only exists in maize plant CNGCs and does not occur in the ion transporters found in other plants, such as rice, *Arabidopsis*, and tomato (*Saand et al., 2015b*; *Nawaz et al., 2014*; *Zelman, Dawe & Berkowitz, 2013*). In this study, 12 ZmCNGCs were identified and the PBCs and hinge domains were highly conserved after alignment, which further confirmed the previous hypothesis (*Saand et al., 2015b*). The conserved motif analysis showed that the motif3 (QWRTWAA[CV]FIQ[AL]AW[RH]RY) pattern was the IQ domain among these 12 ZmCNGCs. The IQ domain is conserved among plant CNGCs and enhances the changeable $Ca^{2+}$-dependent channel control mechanisms in plants (*Fischer et al., 2013*). The results from this study showed that all CNGC proteins contained the IQ motif, which suggested that they bind CaM in a $Ca^{2+}$-dependent manner. Notably, motif4 is the sequence logo of the CNBD domain, which is conserved in most plants and animals (*Jackson, Marshall & Accili, 2007*).

The results showed that the CNGC family in 12 gramineae plants at various evolutionary nodes is adequate for analyzing the phylogeny and evolution of the CNGC gene family in gramineae plants. In Table S1 shows that 4 out of 12 plants under analysis contained <20 CNGC genes. These were *B. distachyon* (16), *O. sativa* (16), *S. bicolor* (13) and maize (12). The ML tree for the 273 CNGCs clearly indicated that gramineae CNGCs clustered into six groups (I, II, III, IV, IVa, and IVb) with significant bootstrap values; Groups IV, IVa and IVb had particularly high bootstrap support values, but group IV CNGCs did not cluster with *B. distachyon*, *O. sativa*, *S. bicolor*, and maize CNGCs. This suggested that there are missing CNGCs that need to be identified or there was duplication during evolution. *Davidson, Guthrie & Lipsick (2013)* showed that gene duplication can lead to the generation of significant numbers of novel genes. Therefore, these results imply that duplication events play a principal role in gene evolution. Phylogenetic, gene structure, and conserved motifs analyses were used to classify the 12 *ZmCNGC* s into five groups with significant bootstrap values. Among these five groups, Group IV *ZmCNGC* s emerged first and had more introns than the genes in the other groups.

### *CNGC* genes play an important role in plant development

The CNGCs are involved in the regulation of plant growth and development (*Chin, Moeder & Yoshioka, 2009b*). This study focused on the role that maize CNGCs play in different tissues by investigating its expression in plant embryo, endosperm and seed expression. The results showed that some *ZmCNGC* genes have tissue-specific expression. For example, *ZmCNGC2*, *ZmCNGC4*, *ZmCNGC6* and *ZmCNGC8* were highly expressed in the roots; *ZmCNGC5* was specifically expressed in pollen; and all Group IVb genes were expressed in the embryo. Group IVb genes are the evolutionary ancestors of CNGC genes and were mainly expressed in the embryo and seeds after pollination, which suggested that Group IVb genes play a significant role in embryo development. In addition, all Group IVb genes were linked to gene duplication and they had a similar expression pattern in the different tissues, which indicated that their function was to enhance adaptability during gene evolution.

Most previous studies have shown that *CNGC* genes are related to pollen development and responses to environmental stimuli. For example, *Arabidopsis CNGC16* is critical for pollen fertility under heat and drought stress (*Tunc-Ozdemir et al., 2013b*), and *CNGC18* has been shown to play a role in pollen tube tip growth (*Frietsch et al., 2007*). In rice, *OsCNGC13* promotes the seed-setting rate by facilitating pollen tube growth in stylar tissues (*Xu et al., 2017*). *ZmCNGC1* and *ZmCNGC5*, two homologous genes of *CNGC16* and *CNGC18*, are mainly expressed in pollen, which indicates that they are predominantly involved in pollen development. Previous studies have shown that *AtCNGC3* is mainly expressed in the embryo, leaves, and roots, and *ZmCNGC4* expression level is similar to that of *AtCNGC3* which was highly expressed during plant development (*Kaplan, Sherman & Fromm, 2007*). These results imply that these genes play crucial roles during the growth and development of maize.

## CONCLUSION

A total of 12 CNGC genes were identified in maize via bioinformatics. The results were based on the presence of a plant CNGC-specific motif that spanned the PBCs and hinge domain in the CNBD of CNGC proteins. Phylogenetic analyses showed that Group IV ZmCNGCs emerged first and had more introns than the other ZmCNGC groups, whereas Groups I and II evolved later. Phylogenetic analysis of 12 gramineae plants revealed that some CNGCs are probably missing or have been duplicated during evolution. Significantly, the *ZmCNGC* genes had diversity in gene structures, protein lengths and sizes. We modified a maize stringent motif (L-X(2)-G-[ED]-ELL-[TSG]-W-[ACY]-L-X(10,20)-[SA]-X-T-X(7)-[EQ]-[AG]-F-X-L) that contained a PBC and a hinge domain to better characterize classification. Expression profiles of the *ZmCNGC* genes were tissue-specific and were related to pollen development. This study provides important information about plant CNGCs during gene evolution.

## ACKNOWLEDGEMENTS

We extend our gratitude to Dr. Lihe Yu for providing valuable suggestions.

### Funding

This work was supported by the Natural Science Foundation of Heilongjiang Province, China (B2016010). The funders had no role in study design, data collection and analysis, decision to publish, or preparation of the manuscript.

### Grant Disclosures

The following grant information was disclosed by the authors:
Natural Science Foundation of Heilongjiang Province, China: B2016010.

### Competing Interests

The authors declare there are no competing interests.

### Author Contributions

- Lidong Hao conceived and designed the experiments, performed the experiments, analyzed the data, contributed reagents/materials/analysis tools, prepared figures and/or tables, authored or reviewed drafts of the paper, approved the final draft.
- Xiuli Qiao conceived and designed the experiments, authored or reviewed drafts of the paper, approved the final draft.

### Data Availability

The information of chromosome distribution of ZmCNGCs and the sequences including DNA sequences, CDS, cDNA, up-stream 1,500 bps of ZmCNGC genes were obtained from BLASTN (the Ensembl Plant database, http://plants.ensembl.org/index.html): accession numbers GRMZM2G148118, GRMZM2G129375, GRMZM2G066269, GRMZM2G023037, GRMZM2G077828, GRMZM2G005791, GRMZM2G068904, GRMZM2G135651, GRMZM2G141642, GRMZM5G858887, GRMZM2G074317, and GRMZM2G090528.

13 maize AKT/KAT genes were downloaded from Ensembl Plant database with gene ID: C234152.1_FGT002,GRMZM2G020859,GRMZM2G081666,GRMZM2G178356, GRMZM2G310569,GRMZM2G156255,GRMZM2G093313, GRMZM2G022915, GRMZM2G171279,GRMZM2G028258,GRMZM2G342375,GRMZM2G428859, GRMZM5G838773,GRMZM2G095861.

The 20 Arabidopsis and 16 rice CNGC protein sequences were obtained from TAIR10 (http://www.arabidopsis.org) and RGAP (http://rice.plantbiology.msu.edu/) respectively. Arabidopsis asscession numbers: AT5G53130, AT3G48010,AT2G23980, AT2G24610, AT5G54250, AT2G28260, AT5G15410, AT1G15990, AT4G30560, AT5G57940, AT3G17700, AT1G01340, AT1G19780, AT2G46450, AT4G30360, AT5G14870, AT4G01010, AT3G17690, AT2G46430, AT2G46440, LOC_Os06g33570, LOC_Os06g33610, LOC_Os03g44440, LOC_Os12g28260, LOC_Os04g55080, LOC_Os02g41710, LOC_Os12g06570, LOC_Os09g38580, LOC_Os02g54760, LOC_Os06g08850, LOC_Os02g53340, LOC_Os06g10580, LOC_Os03g55100, LOC_Os01g57370, LOC_Os05g42250.

## Supplemental Information

Supplemental information for this article can be found online at http://dx.doi.org/10.7717/peerj.5816#supplemental-information.

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
