# Peer review of "Genome-wide identification and analysis of the CNGC gene family in maize"

_PeerJ, doi:10.7717/peerj.5816_

## Round 0.1 · original submission · Major Revisions

In looking over this manuscript I found it a bit difficult to follow due to some of the grammar usage. The form of the grammar would have made it very difficult for a reviewer to understand the science content being proposed. I was able to make some changes which may clear up what I perceived to be the intended statements; however, I did not complete the entire manuscript. I have attached an edited version of your manuscript with suggested changes highlighted so far. I did not necessarily review the abstract, but only tried to make it easier for potential reviewers to read. Please take a look at the manuscript version with suggested revisions and prepare an edited version. Once edited, please re-read it to make sure the message presented is in agreement with yours and re-submit the manuscript so that I may move it forward for review. As a partial review, some of the comments provided are very simply stated with no scientific evaluation provided except for what a software program result might supply. I would recommend a more in-depth evaluation of the results be provided. Thank you for your understanding.

---

## Round 0.2 · Major Revisions

As before, some of the grammatical errors persisted, and as provided by reviewer feedback made salient points hard to understand; a more readable form is still required. Now that the manuscript has had a chance to be reviewed for the science content it appears that there may be some parts of the study that have not reached far enough in the characterization of CNGC genes; a major concern is that this is simply a short computer analysis on new data with routine approaches. There was praise that this work characterizes the genes in a species not yet analyzed, and that adding this to the collection of data may provide additional insights into gene function. However, a major concern is that no experiments were actually planned except that computer analysis was simply done on new data; what was wanted was features identified and tested to perhaps uncover a proposed function; none were identified or proposed. It is recommended that the suggestions provided be undertaken to add significant value to the work. There may be additional research required, but may be quite doable from the perspective of the reviewers. Please address the concerns provided by the reviewers and we hope to see a new revision soon. The reviews were quite severe; however, if you can meet expectations a better outcome can be anticipated. Thank you for your contribution.

·

Basic reporting

see below

Experimental design

see below

Validity of the findings

see below

Additional comments

The author describes first the identification of Cyclic nucleotide gated channels (CNGCs) from maize and subsequently characterized for structural analysis of the protein and other in-silico parameters. The work is unique since the current study reports this family of genes from maize. Yet the manuscript is deficient with many sections. Some of them are mentioned below.
Major points:
a) The work is unique since the current study reports this family of genes from maize. All the 12 ZmCNGC genes should be renamed in this manuscript to better understanding of the following analysis.
b) For identification of the ZmCNGCs, you should concider about potassium AKT /KAT channels (Shaker type) as homologs of CNGCs, due to the presence of transmembrane region, CNBD and ion transport domains and additional ankyrin repeats. Please re-confirm this part, and you’d better to list the conserve domain as a table in this manuscript or in supplementary material.
c) Please re-write the phylogenetic analysis section. To get the accurate model, Maximum likelihood (ML) phylogenetic tree should be used. In the Figure 1, the bootstrap value must be shown in the figure. On the other hand, the branch of group III and group IVb you separated was strange. There is an inverse relationship about the group III and group IVa in this manuscript and the other CNGC family papers. Finally, the annotation of the rice and Arabidopsis CNGCs should be renamed in this figure.
d) In Figure 2, GRMZM2G129375 and GRMZM2G066269 did not show the upstream part in the gene structure part, and the length of those two genes is shorter that other ZmCNGCs. Please confirm the full length and the structure of those two genes.
e) In Figure 3, you should highlight the PBC and hinge region in this figure.
f) CNGC gene families were welled studied in a lot of animals and plants, and no need to do the GO annotation of this gene family. Please remove this part.
g) In Figure 5, Please improve the name of those genes.
h) The expression profiles are important part in this manuscript. Please put the figure s4 into the manuscript. In addition, please explain why the heatmanps of the embryo, endosperm and seed only have 10 ZmCNGC genes, and submit the FPKM date as a supplementary material.
i) Discussion pertaining to conclusion should be made, at many places its mere repetition of introduction or the results. No new hypothesis has been generated, making this part of the manuscript very weak. It very interesting to think about evolutional aspecs of this gene family. The numbers of ZmCNGCs is less than other CNGC gene families (20, 16, 21, 18, and 47 CNGC genes were identified in Arabidopsis, rice , pear, tomato, and wheat). Please discuss more about the Gene duplications and evolutional part, and re-write the conclusion section.

Minor points:
a) there are some grammatical problems (English). I listed some of them here. And I recommend having editorial corrections by a native speaker.
Line 148-152 “Subcellular localization analysis indicated that all of ZmCNGCs localized in the plasma membrane except for GRMZM2G066269 which was localized in the nuclear fraction, this result is consistent with Arabidopsis, for example, previous studies have revealed that CNGCs are majorly localized in the plasma membrane (Lemtiri-Chlieh & Berkowitz 2004).”
Line 182 “is use to evaluate”
Line 222 ”associated with associated with”
b) the figures in this manuscript should be more stringent. Figure 2 must mark the A, B, C if you mentioned as Figure 2A,2B, and 2C. Figure 1,3,5 is unfriendly to understanding.

Reviewer 2 ·

Basic reporting

This manuscript aims to characterize the CNGC family in maize. The manuscript is poorly written (requires English editing) and the authors do not use references properly. The introduction does not cover the literature well. There are many wrong statements or wrong citations associated with statements (see below for some examples).

-References:
Harada et al. 2003 I don't think this paper refers to CNGCs at al
L67 KW et al. 2006 what is that?
L67: Wang et al. 2013 this paper is about guard cells not pollen
Mosher et al. 2010) this paper is about the chimeric CNGC11/12, not CNGC6, 10, 19, or 20

these are just a few examples

Experimental design

There are no experiments. Everything is just computational analysis. There is no research question.

The investigation is shallow with no followup.

Validity of the findings

The novelty of the findings is extremely limited and no experiments were conducted to add any meat to this story.

Furthermore I have doubt about their phylogenetic tree. The authors are very sloppy and mixed up group IVb and group 3 (Maser et al., 2001, Navaz et al., 2014).


The presentation of their “bioinformatics” analysis range from uninteresting to bizarre statements:

-Abstract: Synteny analysis showed that 2, 2 and 1 ZmCNGCs had homologous genes in Sorghum, rice and Brachypodium, respectively

What kind of a statement is that? Of course there is homologs in all other plant species!

-Gene ontology (GO) analysis demonstrated that most of the ZmCNGCs are involved in various biological processes including cellular processes, establishment of localization, and transmembrane transport.

-Is that all they could come up with? A known ion channel gene is membrane localized?

-L161 The physiological and biochemical properties of these 12 ZmCNGC genes are listed in Table 1.
There is no information about any physiological properties.

-L162 The protein lengths ranged from 326
that would indicate either a truncated pseudogene or a wrong sequence. A functional CNGC can not be that short.

-L282: Fig4 is completely uninformative. It does not make sense to do the GO analysis for the whole family. And the conclusion of a “bioinformatics” analysis of a known ion channel family to be involved in transmembrane transport is more than obvious.

-Further, cellular component analysis revealed the localization of ZmCNGCs in the cell and membrane, may be the reason why the subcellular localization of most ZmCNGCs localized in the plasma membrane.
What do the authors want to say here?

-L296 based on the orthology-based predictions following the network in Arabidopsis were constructed.
This seems to be extremely far-fetched and does not provide any useful information

-Suppl 4 seems to be the most interesting. Why it is a supplement? Where is the figure legend?

Additional comments

The authors did not make a lot of effort and did not study about CNGCs at al. The final conclusion is only that maize also has CNGCs, which was obvious from the start.

---

## Round 0.3 · Major Revisions

With an additional reviewer included, most effort was spent trying to read between the lines what was actually trying to be said. The poor use of grammar makes it very difficult to discern the science from the content. I would suggest seeking and working with a professional editing firm fluent in dealing with the English language. There is some good science contained within the manuscript, but it is too challenging to read if the language is not clear. You do not want to present a puzzle to your readers, but a well laid out presentation of the facts which can be well understood and clear. This manuscript is being returned with major revisions expected. In an earlier review it was proposed that the analyses were too simple and simply interpreted results from very few software analyses; basically any laboratory with the same tools might be able to do the same. The requested value-added information just not appear, mainly because of the language barrier. Attention to reviewers comments for suggesting further analyses should be an encouragement to extend your studies further.

Reviewer 2 ·

Basic reporting

There are very serious issues. See point 4

Experimental design

There is only one experiment. I can not comment about the validity of the statements about evolution.

Validity of the findings

The data has to be connected and discussed properly (see point 4)

Additional comments

The revised manuscript by Hao and Qiao has addressed some of the criticisms of the reviewers and I acknowledge that the authors now included their expression data in figure 5. This data is the most interesting in the manuscript and should be analyzed and discussed in much more detail.

The two major problems of the manuscript is a) it appears the authors just do not make much effort to do a thorough analysis of the maize CNGCs. The text is full of platitudes without any significant content.
The second major problem is the English is just not adequate. This may in part be the reason for problem #1. Editing by a service that is specialized in SCIENTIFIC WRITING is required.
In my opinion the manuscript still needs major revisions –especially in writing. There are so many either wrong or badly worded statements. A small selection is in Minor points below.

Besides the writing, major issues for me are:
Fig 2C. What are those motifs??? This looks really strange to me. It is necessary to define those motifs, otherwise the reader can not know which parts CNGC2 and 3 are lacking. Which are the TM domains, the pore and what other motifs are there? If ZmCNGC2 and 3 (and potentially (ZmCNGC9 and 12) are lacking many major domains, then it is questionable whether these are functional channels at all. Or as seen in the Xu et al. 2017 UsCNGC13 paper, CNGCs with truncated C-termini may function as dominant negative isoforms. Also, the fact that no EST hits for ZmCNGC3 were detected could indicate that this I a non-expressed pseudogene.

The results have to be discussed in a more comprehensive manner in the discussion. The data from the phylogenetic analysis (homologs from known Arabidopsis and rice genes) have to be connected to their network analysis (Fig4). What are the genes in the network and how could that connect to the function of the included CNGCs? Why only 5 of the maize CNGCs are in the network?
Most importantly, the data from the expression analysis in Fig 5 needs to be connected and discussed in much more detail. This is the valuable part I see in the manuscript.
Finally, how does the expression and network data connect to the promoter element analysis (SI Fig 4)?

Since the paper is very short on experimental data, at least we can expect a much more thorough analysis of the bioinformatics data presented.

The conclusion should concentrate on some real finding, rather than technicalities like the well-known PBC.

Regarding the expression analysis: Does this mean that ZMCNGC3, 5, and 7 are not expressed in any tissue analyzed?

Minor points/not acceptable statements:

Abstract:
Furthermore, the co-expression network analysis of ZmCNGC genes may establish the importance of better understanding ZmCNGC transduction pathways in maize

Additionally, expression profiles of ZmCNGC genes were shown to express in a tissue-specific pattern

Main text:

Most CNGCs have been characterized by genetic 52 methods and found to be related to plant physiological and molecular functions, including playing 53 vital roles in multiple physiological processes which are involved in signal pathways, plant 54 development, and environmental stresses.

- This sentence does not really have any content, besides the obvious statement that genes have functions.

As a molecular switch, secondary messengers such as cyclic nucleotide monophosphates (cNMPs; 3’,5’-cAMP and 3’,5’-cGMP) and Ca2+/calmodulin (CaM) can regulate 37 CNGCs, those messengers are activated by directing binding of cyclic nucleotides as well as are 38 inhibited by binding of CaM to the CaM binding domain (Saand et al. 2015b; Borsics et al. 2007; 39 Defalco et al. 2016; Kaplan et al. 2007).

- The messengers are not activated by binding, but the CNGCs are activated by binding of the messengers

In plant CNGCs, they are composed of six transmembrane (TM) domains and one pore region 41 between the fifth and sixth TM domains.

- This is not only for plant CNGCs, but all CNGCs (and as reviewer 1 mentioned also AKT/KAT channels). And they also have cytosolic portions at the N and C termini

CNGC16 and CNGC18
56 participate in the pollen development (Tunc-Ozdemir et al. 2013b; Frietsch et al. 2007

- Have to add Gao et al 2016 PNAS

is involved in jasmonic acid induced apoplastic Ca(2+) influx in epidermal cells (Lu et al. 2015).

- I am not sure if Lu et al talk about apoplastic Ca2+. Have to cite Wang et al 2016 for apoplastic Ca2+

Line 59. The Fortuna 2015 paper is talking about flowering time, not abiotic stress.

Subcellular localization analysis showed that all of ZmCNGCs localized in the
154 plasma membrane except for ZmCNGC3 localized in the nuclear fraction

- These are just predictions! The wording sounds like experimental data.

Fig1: Where is AtCNGC3?

Fig2B upstream/downstream refers to UTRs?


182 showed that 2, 2 and 1 ZmCNGCs had homologous genes in Sorghum, rice and Brachypodium, 183 respectively (Table 2).

- The authors mean ZmCNGC1 and 2? Still, this statement is odd. The phylogenetic tree clearly shows many more homologous genes. Do they mean these homologs are only present in maize and sorghum, rice, Brachypodium and maize, not in others?

Suppl Fig 2: I have no idea what this table is supposed to show.

The CNBD identified in CNGCs but no other proteins, 204 hence was recognized as a tool to identify plant CNGCs

-They mean PBC?

Fischer et al. (2013) have showed that IQ as a functional motif within CaMBD and 232 downstream of the CNBD domain, also conserves among plant CNGCs and enhances the 233 changeable of Ca2+-dependent channel control mechanisms in plant.

_ what is changeable?

- cis element analysis: In my opinion, as presented without any additional analysis or experimental validation or any kind of meaningful connection of this prediction to expression data, this analysis is pointless!

-Fig 4. This analysis without further explanation or validation or connection to any other data is pointless. See above. It should be discussed in the context of all other data available, and maybe compared to CNGCs in rice.

Line 297 Compared to other ZmCNGC genes, the gene expression of Group IV genes were more intense

- what is more intense?

suggested that these three ZmCNGC genes obtained gene functions from other plants after 299 duplication.

- They probably mean they obtained novel gene functions that are different from other plants.

These results further illustrated that Group IVb gene play important role in maize 302 gene duplication, evolution and expression.

- What should that mean??? CNGCs have a role for gene duplication?

---

## Round 0.4 · Major Revisions

Again there was some difficulty in interpreting some of the sentence structure. I can see that there is some good science underneath, but it is very difficult to get a clean read on the manuscript. I have attached a scan with some mark-up that you can address for yet another revision. It is not really my role to refine your grammar; it's best that more attention be applied to the science. I can see that the science is there, but the ability to interpret it is lacking. You are getting closer to a good presentation, but I fear the impact of the message is faltering due to progress on other research fronts. There may be minor refinements, but there were too many seen which would classify this as a major revision. Please try and refine your next revision for a better turn-around.

It is strongly recommended that you have a colleague who is proficient in English and familiar with the subject matter, or a professional editing service, review your manuscript.

---

## Round 0.5 · Minor Revisions

The form of the manuscript is very close to what I would have expected in the first draft submission. It was much easier to read and had just a few grammar issues. I have just a few suggested edits; however, it is probably in good enough shape to have reviewers re-read through it again so that they may concentrate more on the science portion. There were a few major issues in the earlier reviews so we should be able to see if the modified manuscript will appease the aforementioned concerns. Please make the suggested edits and I will try for a speedy re-review.

Example of annotation:
LINE NO.: / PREVIOUS FORM / SUGGESTED FORM / [ADDITIONAL NOTES, NONE [.]]

line 57: / vital role in related / vital role related / [remove “in”.]
line 72: / few studies on maize GNGC / few studies on the maize GNGC / [add “the”.]
line 82: / http:// www.arapidosis.org /index.jsp / http:// www.arapidosis.org / [remove “index.jsp”.]
line 87: / threshold of e-value / threshold e-value / [remove “of”.]
line 106: / un-root phylogenetic / un-rooted phylogenetic / [add “ed”.]
line 110: / The tree was and / The tree was / [remove “and”.]
line 118: / motifs analyses / motif analyses / [remove “s”.]
line 134: / high throughput / high-throughput / [add “-”.]
line 159: / the pI values / the calculated pI values / [add “calculated”.]
line 169: / NCBI and / NCBI resource and / [add “resource”.]
line 176: / Figure 1 showed / Figure 1 illustrates / [replace “illustrates”.]
line 202: / expansion occurred / expansion appeared to have occurred / [.]
line 218: / genes emerged / genes may have emerged / [add “may have”.]
line 224: / investigating, the genome / investigating the genome / [.]
line 231: / strongly purifying / ? / [you may want to re-word.]
line 235: / motif of / ? / [missing information.]
line 246: / various patterns / ? / [something more descriptive.]
line 312: / had no ESTs and / had no EST alignments and / [.]
line 332: / Supplemental File 5: Table 1 / In Supplemental File 5; Table 1 /[.]
line 335: / bootstrap values Groups IV / bootstrap values; Groups IV /[.]
line 343: / was then Group I / appears to have contained / [.]
line 350: / genes has tissue-specific / genes have tissue-specific / [.]
line 356: / that there function / that their function / [.]
line 376: / had diverse in gene / had diversity in gene / [.]
line 379: / hinge domain. / hinge domain to better characterize classification. /[.]

---

## Round 0.6 · accepted · Accept

Thank you for providing the latest suggested edits. In looking over the previous suggestions, I will recommend moving this forward rather than adding another review process.

I only suggest two changes in this version; at line 98 add the word “were” to change to “DNA sequences were obtained”, and at line 128 change the case of “NEW” to new”. The original name of the database was called PLACE, and the website provides a newer version.

Otherwise the manuscript appears in good shape and should be ready to move forward. I apologize for this lengthy process; however, it is important that the content be in the best condition as possible to speed the review process. In the future, it would help to have a professional editing service fluent in the English language to provide assistance in preparing the manuscript.

On the science side, the identification of new CNGC genes in maize will be useful to others comparing this gene family amongst different species, and may lead to further characterization of the diversity in function. Thank you for your contribution; consider it stamped accepted for publication.

#